# Unlocking Long-Horizon Agentic Search with Large-Scale End-to-End RL

**Jiaxuan Gao**[12]**, Wei Fu**[12]**, Minyang Xie**[1]**, Shusheng Xu**[2]**,**
**Chuyi He**[2]**, Zhiyu Mei**[2]**, Banghua Zhu**[3]**, Yi Wu**[1*]

[1] IIIS, Tsinghua University, [2] Ant Group, AReaL Team
[3] University of Washington
samjia2000@gmail.com, jxwuyi@gmail.com

## Abstract

Recent advancements in LLM-based agents have demonstrated remarkable capabilities in handling knowledge-intensive tasks using external tools. One representative example is *search agent*. Existing open-source search agents heavily rely on advanced commercial LLMs: they either collect trajectories from the larger, stronger models for supervised fine-tuning or directly use them as specialized tools. In this work, we develop *ASearcher*, a *single-model* search agent purely trained by reinforcement learning (RL) *without using any commercial APIs for data or tools*. Based on an RL-trained QwQ-32B model, *ASearcher* is capable of conducting complex reasoning, such as uncertainty analysis and conflict verification, and achieves comparable performances to commercial search agents. There are two key techniques to unlock such long-horizon information-seeking abilities: first, we design a two-staged agentic process to synthesize high-quality QA pairs as the training data for RL; second, we conduct large-scale *long-horizon* RL, allowing the agent to take up to 128 actions per rollout for sufficient exploration. In particular, after RL training, *ASearcher* achieved scores of GAIA 58.1, xBench 51.1, and Frames 74.5 using only basic search tools. Furthermore, *ASearcher* also demonstrates strong zero-shot transferability: *ASearcher* can be further augmented with an additional summary tool, which is supported by DeepSeek-V3, and test-time scaling, which aggregates the answer from 16 parallel rollouts. With both zero-shot enhancements, the performances of *ASearcher* further rise to 71.8, 75.0, and 83.4, respectively, outperforming OpenAI DeepResearch and Kimi-Researcher, suggesting the great potential of RL scaling for agentic tasks. We release all the code and data at anonymous link. The model will be released after the review process.

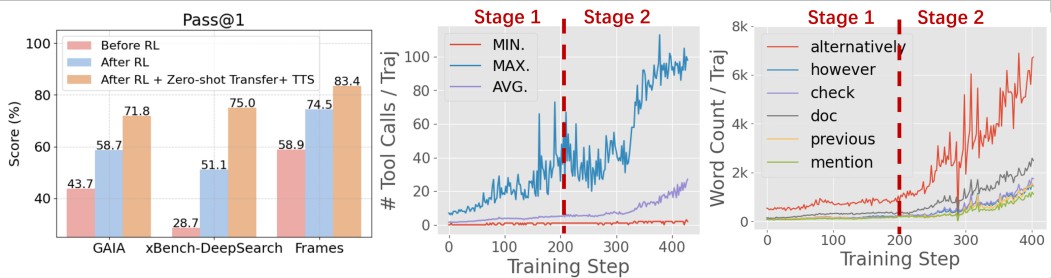

Figure 1: **(Left) End-to-end RL brings substantial improvements to a simple agent:** Through RL training, our agent, *ASearcher*, obtains +15.0, +22.4, and +15.6 improvements on GAIA, xBench, and Frames, respectively. **(Middle) During RL training, *ASearcher* learns to conduct long-horizon search**, with tool calls progressing from an average of only 1.67 initially to over 20 tool calls in latter training stages. **(Right)** Count of keywords during the training process reveals **emergence of complex search behaviors** including reflective behaviors and referencing external information. Our detailed case study in Appendix B also shows that the agent **learns expert-level search strategies**.

---

* Corresponding author

# 1 INTRODUCTION

Recent advances in LLM-based agents have demonstrated remarkable capabilities in solving complex, knowledge-intensive problems by leveraging single or multiple external tools (Xi et al., 2025; Yao et al., 2023; Wang et al., 2024a). Among the diverse capabilities, **deep information retrieval** using tools stands out as a particularly critical aspect of advanced search agents (OpenAI, 2025; Google Team, 2025; Perplexity Team, 2025). For a concrete example, a seemingly simple question like *"How many gold medals did China win at the 2012 London Olympics?"* actually requires intricate reasoning. At the time, China was credited with 38 golds, but a decade later, two doping disqualifications in women's race walking led to an additional gold medal to China, raising the total to 39. This illustrates how search agents must reconcile historical records with noisy, sometimes conflicting, information from diverse sources and identify the underlying causes for the conflicts to deliver accurate answers.

To equip agents with this deep retrieval capability, recent open-source approaches frequently depend on commercial LLMs, either to generate expert data or to serve as specialized sub-modules within a complex multi-model framework (Li et al., 2025a; Tao et al., 2025; Li et al., 2025b; Team, 2025a). For instance, AFM (Li et al., 2025b) collects supervised fine-tuning data from a multi-agent framework powered by multiple advanced commercial models such as Claude-Sonnet-4 (Anthropic, 2025) and Gemini-2.5-Pro (Google Team, 2025). On the other hand, MiroThinker (Team, 2025a) employs distinct commercial LLMs and VLMs for specialized tasks such as audio transcription, visual question answering, and complex reasoning. This reliance on proprietary models raises a fundamental question: *can we achieve the performance of commercial systems without dependence on commercial models?*

In this work, we present *ASearcher*, a search agent trained solely by Reinforcement Learning (RL), and show that *purely end-to-end RL can enable the emergence of advanced long-horizon search strategies, despite that* ASearcher *is based on a single model using only search tools.*Particularly, two techniques are the key to the advanced information-seeking abilities in *ASearcher*. First, we develop a scalable QA synthesis agent to generate a high-quality dataset of 25.6k challenging QA pairs that necessitate multi-turn tool use. In training time, we employ a two-stage curriculum that progressively focuses on challenging tasks. Specifically, the agent is initially trained over questions spanning diverse difficulties, including those easy questions requiring only one or two tool calls. After the agent learns preliminary search capabilities, we shift the training distribution to focus on long-horizon tasks that require a minimum of five tool calls. Second, we train the agent with large-scale long-horizon RL with a large turn limit of 128 per rollout trajectory. A large turn limit encourages the exploration and discovery of sophisticated, long-horizon strategies. To ensure high training efficiency, we employ fully asynchronous agentic RL training based on AReaL (Fu et al., 2025) that decouples trajectory collection from weight updates for training efficiency.

We use a large reasoning model QwQ-32B (Team, 2025b) as the base model in our experiments. During RL training, our agent, *ASearcher*, learns to conduct significantly more complex searches, with an increasing average number of tool calls from only 1.67 calls at the beginning to more than 20 calls. A detailed case study and keyword analysis further reveal the emergence of complex search behaviors, such as conducting uncertainty analysis and verification searches. Our finding on search agents is akin to DeepSeek-R1 (DeepSeek-AI et al., 2025), where the emergent reasoning capabilities can be fully incentivized by RL. We evaluate our agents on challenging benchmarks, including GAIA (Mialon et al., 2023) , xBench-DeepSearch (Xbench-Team, 2025), and Frames (Krishna et al., 2024). With only a single model and basic search tools, *ASearcher* achieves competitive scores of 58.7, 51.1, and 74.5 (Avg@4) on GAIA, xBench, and Frames, respectively, demonstrating that strong performance is attainable with a single-model design. Finally, *ASearcher* also demonstrates strong zero-shot transfer ability to external summary tools. By employing an external summary tool supported by DeepSeek-V3 and applying test-time scaling techniques, where we aggregate the conclusions from 16 indepdent runs, the performance can be further enhanced, rising to 71.8, 75.0, and 83.4, respectively, and achieving results competitive with commercial systems.

# 2 RELATED WORKS

**Search Agents.** Some works have investigated agent workflows to leverage tools for solving complex tasks (Li et al., 2025c; Zhao et al., 2025). Prompt-based methods, though effective for rapid development, are fundamentally limited by the capacity of the underlying LLMs. Some works

attempt to construct SFT trajectories for LLMs. For instance, Asai et al. (2023); Yu et al. (2024) leverage LLMs to synthesize trajectories for SFT. Prior works investigate Reinforcement learning (RL) methods to enhance the LLM-based agents, mostly focusing on multi-hop QA benchmarks. Jin et al. (2025); Song et al. (2025); Chen et al. (2025); Zheng et al. (2025) perform RL training with multi-hop QA data and observe an increasing amount of tool calls. More recently, researchers focus on deep research tasks, by fine-tuning complex prompt-based agents through offline RL (Li et al., 2025d), SFT on diverse trajectories (Sun* et al., 2025; Li et al., 2025a; Team, 2025a), adopting multi-agent framework (Li et al., 2025b), and constructing challenging QAs for RL training. (Tao et al., 2025). In this work, we focus on deep information retrieval, and show that RL alone can equip a single-model agent with advanced long-horizon search capabilities.

**Agentic Reinforcement Learning.** Recent works have begun to improve the agentic capabilities of LLMs and LRMs through online Reinforcement Learning. Prior works have investigated agentic RL in various domains, such as search agents (Jin et al., 2025; Li et al., 2025a;d; Tao et al., 2025) and coding (Luo et al., 2025a; Wei et al., 2025). A critical aspect of agentic RL is to activate and enhance the tool-calling capability of the models. Wei et al. (2025) and Luo et al. (2025a) investigate training coding agents for resolving real-world engineering questions. More recently, Li et al. (2025a); Tao et al. (2025); Li et al. (2025b); Team (2025a) investigate training deep search agents by first running Supervised Fine-Tuning to equip the agent with basic long-horizon search capability as cold-start and then running Reinforcement Learning to further enhance the agent. In this work, we focus on search agents and show that a simple agent can learn complex long-horizon search strategies through reinforcement learning.

## 3 ASearcher

In this work, we present *ASearcher*, which unlocks search intelligence in search agents through large-scale asynchronous RL training. In the subsequent sections, we present the agent design, the training data as well as data synthesis agent, and fully asynchronous reinforcement learning training.

### 3.1 AGENT DESIGN: ONE SINGLE MODEL FOR ALL

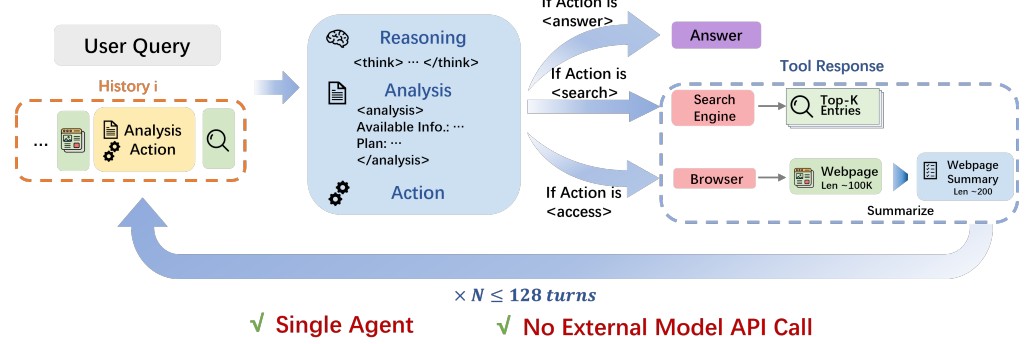

Figure 2: *ASearcher* utilizes a simple agent design with two basic tools including search and browsing tools, without relying on any external models. The agent is capable of both reasoning and summarizing lengthy web contents.

We employ a simple agent design in *ASearcher*, as illustrated in Fig. 2.

**Agent Input.** In each turn, the search agent takes in the user query as well as the history of resolving the user query. The history contains previous tool responses including search results and webpage summaries, and also previous analysis and history actions.

**Reasoning, Analysis, and Action.** In each turn, given the user query and history context, the agent generates three components to further conduct in-depth analysis and exploration,

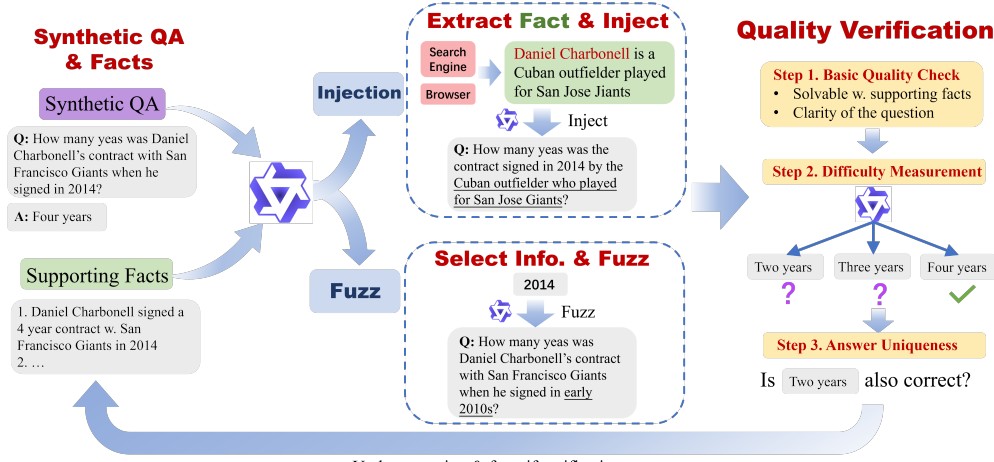

Figure 3: Data Synthesis Agent. Starting from a seed QA, the data synthesis agent iteratively modifies the question through two actions, *Injection* and *Fuzz*.

- **Reasoning:** In this part, the agent conducts internal reasoning over the current situation. Since we instantiate the agent with large reasoning models such as QwQ-32B, generating reasoning process is naturally supported by the underlying model. In the reasoning process, the agent analyzes available information, evaluates the query resolving progress, reflects over previous results, determines the unresolved aspects, and deduces concrete plans for future turns. Note that the reasoning part is only used for guiding the generation of subsequent analysis and action. Since this reasoning part usually contain noisy and lengthy model-generated texts, the reasoning part is not included in future history to ensure a clean history.

- **Analysis:** The analysis part is a summarization of the reasoning part, where the agent extracts the key conclusions derived from the reasoning process and also makes a plan for the subsequent step.

- **Action:** After thoroughly analyzing the current state with the reasoning and analysis parts, the agent finally determines the next-step action. The agent could either answer the question and terminate the execution process, or invoking external tools to obtain new information from external sources.

**Tools.** When the agent determines to invoke external tools, two basic tools are available: **a search engine** and **a web browser**. When the action is "<search>", the search engine takes a query as input and returns relevant snippets along with corresponding URLs. When the action is "<access>", the web browser accepts an URL and returns content of the webpage.

**Webpage Summarization.** Note that real-world webpages are usually very long, easily exceed 32K tokens. Therefore, we split the webpage into several chunks, with a maximum character count of 10k per chunk. We employ the agent to summarize each chunk into a compact summary.

## 3.2 TRAINING DATA

Our training data are from two primary sources, including samples filtered from open-source datasets and synthetic high-quality question-answer (QA) pairs.

**Open-source Data.** We begin with the training sets of HotpotQA (Yang et al., 2018) and 2Wiki-MultiHopQA (Ho et al., 2020). We employ a model-based filtering process. We first train a model on the full set of open-source data with RL fllowing Jin et al. (2025), and then generate 16 responses for each question using the trained model. Finally, we filter out questions that are too hard for the model or too easy for the model. Finally, from a total of 304k QA pairs, we retain 16k challenging samples.

**Data Synthesis Agent.** We further develop a data synthesis agent supported by QwQ-32B to create high-quality question-answer pairs. As shown in Fig. 3, the data synthesis agent begins with a seed

question, and iteratively modifies the question to increase the complexity. To ensure the synthetic question is strictly aligned with reliable sources, a list of *supporting facts* obtained during the question synthesis process is continuously updated. At each step, the agent automatically selects between two key actions,

- **Action 1: Injection** aims to enrich the context of the question by inserting facts related to the question. The agent first selects an entity in the question and then obtains one piece of related fact about the selected entity from external sources such as Wikipedia. Then a new question is proposed by *injecting* the fact into the question.
- **Action 2: Fuzzing** blurs certain details in the question to increase the uncertainty level of the question. For example, "Catskill Mountain Railroad" could be replaced with "a historic mountain railway".

To ensure that a synthetic question is of high quality and to precisely evaluate the difficulty, we incorporate a rigorous *quality verification* phase for assessing synthetic questions. This verification phase includes three steps: *basic quality check* that assess the clarity and resolvability of the question, *difficulty measurement* by employing QwQ-32B to direct generate answers without tools, and *answer uniqueness check* by evaluating whether any of the mismatched answers generated during the Difficulty Measurement step could serve as alternative valid answers.

Through iterative injection and fuzzing, the data synthesis agent produces questions that involve complex information and high uncertainty, requiring extensive search and reasoning to find the correct answer. After completing the question synthesis process, we filter out questions that the LRM can directly generate the correct answer without using tools.

**Two-Stage Curriculum.** During training time, we employ a two-stage training data scheme. In the first stage, we apply RL training on the full training set, which include QAs spanning different difficulties. This wide range of training distribution trains the agent to equip basic tool-calling and reasoning capabilities. In the second stage, to further activate the long-horizon search capability of the agent, we remove QAs that are solvable with less than 5 tool calls and use the rest data for the second training stage.

### 3.3 ASYNCHRONOUS AGENTIC RL TRAINING

#### 3.3.1 EFFICIENCY CHALLENGE OF LONG-HORIZON AGENTIC RL

**High Variance in Trajectory Execution Time.** During RL training, we use a large turn limit of 128. In practice, when using a large turn limit at training time, long trajectories introduce significant variance in execution time. We first analyze the number of tool calls during RL training of our QwQ agent (Fig. 1) and observe that the longest trajectories can span dozens more tool calls than shorter ones. Second, we also report the total number of output tokens per trajectory during training in Fig. 4. As illustrated in the figure, the training process involves extremely long trajectories. The data distribution reveals that these lengthy trajectories constitute a very small proportion of the samples. This disparity could lead to highly unpredictable per-trajectory runtime, complicating training efficiency.

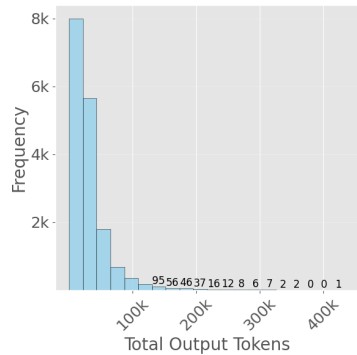

Figure 4: Distribution of the output lengths of trajectories generated during RL training (recorded from Step 290 to Step 310).

**Efficiency Issues of Agentic RL Training.** Both prolonged execution and high runtime variance degrade RL training efficiency. We take one-step-off RL training system (Luo et al., 2025b) as a representative example for batch generation RL systems. As shown in Fig. 5, though this system overlaps trajectory rollouts with model training, batch generation remains bottlenecked by the slowest trajectory (e.g., trajectory 7), causing GPU idle time and under-utilization.

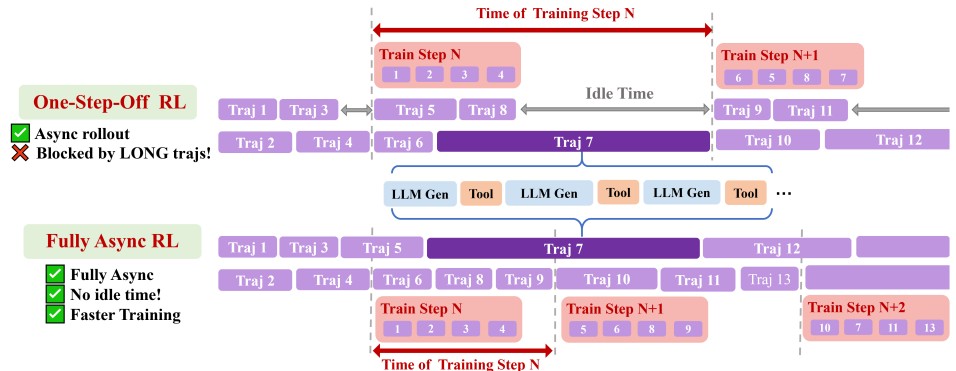

Figure 5: One-Step-off RL v.s. Fully Asynchronous RL. In batch generation systems, a batch should wait for the longest trajectory, leading to significant GPU idle time. In contrast, fully asynchronous RL achieves faster training than batch generation RL by fully decoupling training and trajectory generation, achieving near-full resource utilization for trajectory generation.

### 3.3.2 FULLY ASYNCHRONOUS RL TRAINING.

To ensure efficient agentic RL training, we adopt a fully asynchronous training paradigm. Notably, our approach incorporates asynchornization at the two distinct aspects.

**Asynchronous Trajectory Rollouts.** Trajectory rollouts are collected in parallel and do not directly interfere with each other. Each trajectory independently sends tool calling requests to corresponding servers and LLM generation requests to the LLM inference engine. Concurrent requests from different trajectories are automatically handled by the servers. Fully independent trajectory execution ensures a trajectory does not need to wait for other trajectories when generating LLM responses and waiting for tool calling responses, thereby improving training efficiency.

**Decoupled Rollout and Training.** Besides asynchronous rollout, trajectory rollouts and model updates are also fully decoupled. In Fig. 5, we compare our fully asynchronous RL training with one-step-off RL training, which utilizes asynchronous rollout within batches. In fully asynchronous RL training, long trajectories do not block generation and can span multiple versions, significantly reducing GPU idle time and achieving near-full GPU utilization during generation. On the training side, a training step is launched as soon as sufficient trajectories are collected to form a batch.

### 3.4 TRAINING DETAILS

**MDP Formulation.** We follow the formulation of Markov Decision Process (MDP). Formally, an MDP is defined by the tuple $(S, A, T, R)$. Here $S$ represents the state space, usually containing the history, search results, and retrieved webpages. $A$ denotes the action space and an action includes tokens generated by the agent. Some tool calling could be extracted from the action through specific tags, e.g. <search> search query </search>. $T(s'|s, a)$ is the transition function. At each timestep, the agent receives a state $s_t$ and generates an action $a_t$ with policy $\pi : S \to A$. The goal of the agent is to maximize the return $J(\pi) = \mathbb{E}\left[\sum_{t=0}^{\infty} R(s_t, a_t)\middle| a_t \sim \pi(s_t)\right]$.

**GRPO Training.** We employ the GRPO (Shao et al., 2024) algorithm to train search agents. Specifically, for each input question $x$, $G$ trajectories $\tau_1, \tau_2, \cdots, \tau_G$ are generated where $\tau_i = (s_0^i, a_0^i, s_1^i, \cdots, s_{T_i}^i)$. To optimize the agent, we employ the following loss,

$$\mathcal{J}_{GRPO}(\theta) = \mathbb{E}_{x\sim\mathcal{D}, \{\tau_i\}_{i=1}^{G}\sim\pi_{\theta_{old}}(\cdot|x)}\left[\frac{1}{G}\sum_{i=1}^{G}\frac{1}{\sum_{t=0}^{T_i-1}|a_t^i|}\sum_{t=0}^{T_i-1}\sum_{j=1}^{|a_t^i|}\min\left(\frac{\pi_\theta(a_{t,j}^i|s_t, a_{t,<j}^i)}{\pi_{\theta_{old}}(a_{t,j}^i|s_t, a_{t,<j}^i)}\hat{A}_i,\right.\right.$$
$$\left.\left.\text{clip}\left(\frac{\pi_\theta(a_{t,j}^i|s_t, a_{t,<j}^i)}{\pi_{\theta_{old}}(a_{t,j}^i|s_t, a_{t,<j}^i)}, 1 - \epsilon, 1 + \epsilon\right)\hat{A}_i\right)\right] \tag{1}$$

where $\epsilon$ is a hyperparameter, and $\hat{A}_i$ is the advantage for the $i$-th trajectory, computed based on the relative rewards of all trajectories within each group.

**Dynamic Filtering.** To enhance training efficiency, we implement dynamic filtering to exclude queries that lack meaningful training signals. Specifically, we remove queries where all responses yield identical rewards (resulting in zero advantages).

**Reward Function.** We adopt a sparse-reward setting where rewards are computed at trajectory completion. For reward function, we utilize LLM-as-Judge(Liu et al., 2023; Wang et al., 2024b) as the reward function and omit format rewards, as large reasoning models such as QwQ-32B could inherently maintain proper output formatting with high probability.

## 4 EXPERIMENTS

### 4.1 EXPERIMENT SETUP

**Benchmarks.** We conduct evaluation on a suite of challenging benchmarks, including Frames (Krishna et al., 2024), GAIA (Mialon et al., 2023), xBench-DeepSearch (Xbench-Team, 2025), and HLE(Li et al., 2025d). Frames contains 824 challenging questions for evaluating the ability of the agent to synthesize accurate responses from multiple sources. GAIA contains real-world questions that demand multi-turn tool calls and step-by-step problem solving. xBench-DeepSearch consists of 100 challenging Chinese questions constructed by human experts, evaluating the agent's in-depth planning and reasoning capabilities. HLE (Human's Last Exam) features expert-level difficulty across a wide range of disciplines, not only requiring the agent to search for related materials, but also understanding and solving domain-specific questions. For GAIA, we use the 103 examples from the text-only validation subset (Li et al., 2025c). For HLE, we use the 500-size subset following Li et al. (2025d).

**Baselines.** We compare *ASearcher* against different sets of baselines:

- **Commercial Deep Research Agents.** We make a comparison with OpenAI DeepResearch (OpenAI, 2025) and Kimi-Researcher (MoonshotAI, 2025).

- **General LLMs using Tools.** We evaluates general LLMs equipped with external tools including Qwen3-30B-A3B, Qwen3-235B-A22B (Yang et al., 2025), OpenAI-o3 (OpenAI, 2025), DeepSeek-R1 (DeepSeek-AI et al., 2025), and Claude-4-Sonnet (Anthropic, 2025).

- **Open-source Search Agents.** Finally, we make comparison with a set of 32B-scale open-source search agents, including Search-o1(QwQ-32B) (Li et al., 2025c), Search-R1-32B (Jin et al., 2025), WebThinker-QwQ (Li et al., 2025d),SimpleDeepSearcher-QwQ (Sun* et al., 2025) and WebDancer-32B (Wu et al., 2025), WebSailor-32B (Li et al., 2025a), and WebShaper-32B (Tao et al., 2025). We also include AFM-RL-32B (Li et al., 2025b), that adopts a multi-agent design, and MiroThinker-32B-DPO (Team, 2025a), that utilizes more tools beyond search tools, including tools for image and audio processing.

**Evaluation Metrics.** We adopt LLM-as-Judge (LasJ) as the main metric for evaluating the performance. For LLM-as-Judge, an LLM (Qwen2.5-72B-Instruct) is prompted to assess the correctness of outputs. For *ASearcher*, We report pass@1 score by evaluating 4 seeds. For baselines, we report the scores reported in official reports if there are any.

**Training Details of *ASearcher*.** We set the turn limit as 128 and the batch size is set as 64 for *ASearcher*. We use AdamW optimizer with a learning rate of 2e-6. Our training framework is built up on AReaL (Fu et al., 2025). Training of *ASearcher* takes approximately 16k H800 GPU hours.

### 4.2 MAIN RESULTS

**Benchmark Performance.** Table 1 shows experiment results on challenging QA tasks that require advanced and search strategies. Our agent, *ASearcher*, achieves competitive Pass@1 scores on GAIA, Frames, and HLE, outperforming a wide range of previous 32B-scale agents. Notably, *ASearcher*

Table 1: Pass@1 results of *ASearcher* and baselines. [†] indicates results are obtained from official reports. For open-source search agents, we use "No Commercial LLM" to indicate that the agent does not use commercial models as a component of agent design or for data collection, and "Non-Search Tools" to indicate the tools used by the agent besides basic search tools. *ASearcher* outperforms a wide range of 32B-scale open-source agents, with single model and no extra tools. When integrating external models as summary tool and applying a test-time scaling approach, *ASearcher* is able to achieve on-par performance with commercial systems.

| Method | No Commercial LLM | Non-Search Tools | GAIA | xBench-DeepSearch | Frames | HLE |
|---|---|---|---|---|---|---|
| **Commercial Deep Research Agents** | | | | | | |
| Kimi-Researcher | - | - | - | 69.0[†] | 78.8[†] | 26.9[†] |
| OpenAI DeepResearch | - | - | 67.0[†] | - | - | 26.6[†] |
| **General LLMs using Tools** | | | | | | |
| OpenAI-o3 | - | - | 70.5[†] | 66.7 [†] | 84.0[†] | 20.2[†] |
| Qwen3-30B-A3B | - | - | 35.9[†] | 32.0[†] | 56.4[†] | 13.2[†] |
| Qwen3-235B-A22B | - | - | 45.6[†] | 46.0[†] | - | 20.0[†] |
| DeepSeek-R1 | - | - | - | 55.0[†] | 82.0[†] | 24.8[†] |
| Claude-4-Sonnet | - | - | 68.3[†] | 64.6[†] | 80.7[†] | 20.3[†] |
| **Open-source Search Agents** | | | | | | |
| Search-o1 (QwQ) | ✓ | - | 48.1 | 40.3 | 63.6 | - |
| Search-R1-32B | ✓ | - | 28.6 | 19.5 | 44.1 | - |
| WebThinker-QwQ | ✓ | - | 48.5[†] | 32.8 | 57.7 | 15.8[†] |
| Simple DS-QwQ | ✓ | - | 50.5[†] | 35.8 | 68.8[†] | - |
| WebDancer-QwQ | ✗ | - | 51.5[†] | 40.0 | 63.8 | - |
| WebSailor-32B | ✓ | - | 53.2[†] | **53.3**[†] | 69.8[†] | - |
| WebShaper-32B | ✓ | - | 53.3[†] | - | - | - |
| AFM-RL-32B | ✗ | Code Sandbox | 55.3[†] | - | - | 18.0[†] |
| MiroThinker-32B-DPO | ✗ | Visual Question Answering, Audio Transcription, Linux Sandbox | 60.9[†] | 56.0[†] | 74.8[†] | 20.6[†] |
| **Ours (QwQ-32B)** | | | | | | |
| *ASearcher* | ✓ | - | 58.7 | 51.1 | 74.5 | 21.5 |
| + Summary=DeepSeek-V3 | ✗ | - | 60.3 | 56.4 | 76.6 | 23.4 |
| + Test-time Search (K=16) | ✗ | - | 71.8 | 75.0 | 83.4 | 24.6 |

even achieves on-par performance as MiroTinker-32B-DPO that uses extra tools besides search tools. These results further highlight the superiority in handling long-horizon and real-world tool use.

**Zero-Shot Transfer with Summary Tool & Test-time Scaling.** Although we adopt single-model during training time, the agent design of *ASearcher* is generalizable. Specifically, the webpage summarization process could be supported by external models. We here use DeepSeek-V3 as the webpage summarization model. From Table 1, it is clear that using a more powerful model for summarization improves the performance of *ASearcher*. We further investigate test-time scaling. Specifically, we run $K = 16$ independent runs for each problem in the test set and aggregate the conclusions from these independent runs with DeepSeek-V3. As shown in Table 1, test-time scaling approach leads to competitive performance with the commercial agents, including Kimi-Researcher, OpenAI DeepResearch, and OpenAI o3. Please refer to Appendix. E for additional results and implementation details of Test-time Scaling.

## 4.3 EMERGENT BEHAVIORS

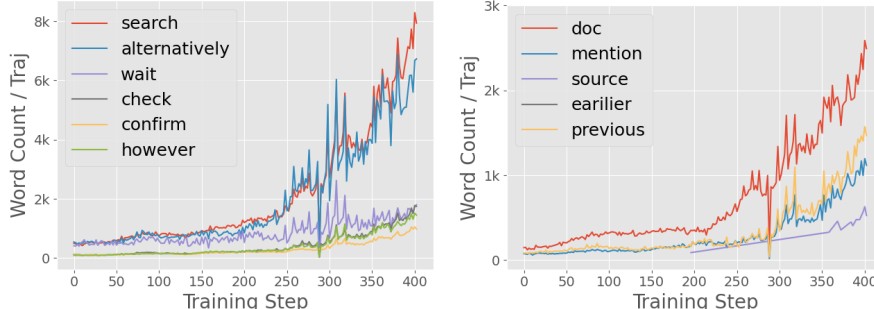

Figure 6: Left: Word count of reflective keywords during training time. Right: Word count of keywords indicating explicit reference of external information.

**Keyword Analysis.** In Fig. 6, we plot the word count of different keywords during the training process. These keywords include words that indicate reflection behaviors of the agent, such as "alternatively", "however", and "wait", and also words indicating that the agent is referencing to external information, such as "doc" and "mention". From Fig. 6, it is clear that the agent learns to reflect over previous actions and conclusions. Interestingly, we also see that, in the second stage of RL training where the training focuses on challenging tasks, the agent learns to refer to external information, as evidenced by the increment in the word count of "doc" and "mention" after step 200.

**Case Study** In Fig. 7, we provide a detailed case study on an extremely challenging question from GAIA (Mialon et al., 2023). Specifically, we analyze Search-R1-32B (Jin et al., 2025) and *ASearcher*. The detailed trajectories are provided in Appendix B. In this question, to identify the correct answer, the search agent should first find out the mentioned species according to condition "genus named for Copenhagen", identify the correct 2021 article based on the citation in the wikipedia page of the species, and then find out the papers of the two mentioned persons.

In Fig. 7, Search-R1-32B is unable to decompose the complex query into individual components, consequently only making ambiguous queries that involve too many unknown information. The agent also has severe hallucinations, producing conclusions that are not supported by the search results. Finally, it fails to resolve all unknown information. In contrast, *ASearcher* decomposes the complex query into precise queries. *ASearcher* could also summarize all related information from a webpage and analyze all candidate answers. Finally, after the correct answer "Mice" is found, the agent spends further turns on verifying previous conclusions before reporting the final answer.

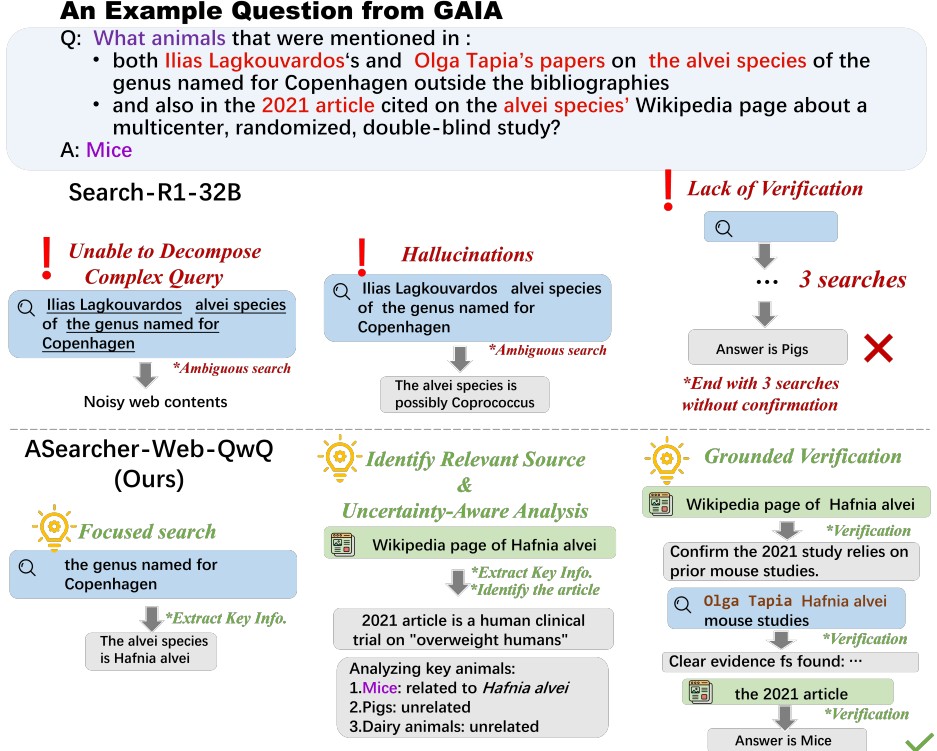

Figure 7: A case study on a complex query from GAIA. **Search-R1-32B** is unable to break down the complex question and has severe hallucinations. It is also worth noting that, since the turn limit is set as a small value, e.g. 4, during training, the model only exhibits a short tool-use horizon. Our end-to-end RL agent, *ASearcher*, exhibits key behaviors featuring Search Intelligence: *uncertainty-aware reasoning* (list and examine candidate answers), *precise extraction* from noisy contents, and *grounded verification*.

## 4.4 ABLATION STUDY

We perform ablation studies over various key components of *ASearcher*. Specifically, we aim to answer the following questions,

- **Q1:** How does the training-time turn limit influence the final performance and learned behaviors?
- **Q2:** Is the two-stage curriculum necessary for activating the long-horizon search capabilities?
- **Q3:** How does the training data of *ASearcher* compare with training data of baseline?

**Experiment Setup.** To carry out the ablation studies, we perform RL training starting from the Stage 1 checkpoint of *ASearcher*. All ablation studies are trained for 200 steps to ensure the same number of training steps as the full training recipe of *ASearcher*. The ablation studies are evaluated on GAIA (Mialon et al., 2023) and xBench-DeepSearch (Xbench-Team, 2025).

Table 2: Ablation Study on Training-time Turn Limit and Data Quality. The ablation studies reveal that both *a large turn limit* and *high-quality* training data are the key to unlocking the long-horizon search capability of the model.

|  | # of Tool Calls at Training Time | GAIA | xBench-DeepSearch |
|---|---|---|---|
| ASearcher (Full) | 26.59 | 58.7 | 51.1 |
| **Ablating Training-time Turn Limit** | | | |
| ASearcher w. Turn Limit=10 | 3.48 | 49.2 | 39.3 |
| **Ablating Training Data** | | | |
| ASearcher w. Stage 1 Data Only | 5.40 | 51.6 | 43.0 |
| ASearcher w. AFM Data | 4.12 | 50.9 | 39.9 |

The results of ablation study is as shown in Table. 2. We highlight key conclusions derived from our ablation analysis,

- **(Q1) Training-time Turn Limit.** A large turn limit is crucial for enabling the model's long-horizon search capability. Specifically, training with a large turn limit significantly surpasses training with a small turn limit (e.g., Turn Limit = 10). With a large turn limit, the agent is provided with a rich exploration space for complex search strategies.
- **(Q2) Two-Stage Curriculum.** Compared with continued training using only Stage 1 data, the full training recipe incorporating a two-stage curriculum helps the agent to learn stable long-horizon search by focusing on challenging queries that necessitate at least five tool calls. This shows the importance of the progressive curriculum for mastering complex tasks.
- **(Q3) Data Quality.** We also compare our training data with that of a concurrent work, **AFM (Li et al., 2025b)**, by applying asynchronous RL training with a 128-turn limit using the AFM training data. Training with AFM data is unable to incentivize complex search capabilities and achieves sub-optimal benchmark performance, highlighting the high quality and complexity of our synthesis data.

## 5 CONCLUSION

In this work, we present *ASearcher*, investigating large-scale RL training for search agents. Our contribution includes a fully asynchronous agentic RL training system and a data synthesis agent for large-scale high-quality QA construction. Through large-scale RL training, *ASearcher* demonstrates emergent complex strategies using only a single model and basic search tools. We hope our work could benefit future work on training advanced agents for a broader range of applications.

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

## A   LLM USAGE

In the development process of this work, LLM were used in the writing and polishing of the manuscript. Specifically, we used an LLM to assist in refining the language, improving readability, and ensuring clarity in various sections of the paper. The LLM helped with tasks such as sentence rephrasing, latex formatting, and grammar checking.

## B   FULL CASE STUDY

In this section, we provide a detailed case study on an extremely challenging question from GAIA (Mialon et al., 2023). Specifically, we analyze Search-R1-32B (Jin et al., 2025) and Search-o1 (QwQ) (Li et al., 2025c) in Fig. 8.

**Solution Path of the Sample Question.**   In Fig. 8, our case study is carried out on a question requiring finding some specific animal given 2 **conditions** and 4 **unknown variables**. To identify the correct answer, the search agent should first find out the mentioned species **U1** according to condition **C1**, identify the correct article **U2** that satisfies condition **C2**, and then find out the papers listed in **U3.1** and **U3.2**. Finally, the correct answer should be determined by cross referencing the article **U2** and the papers **U3.1**&**U3.2**. To summarize, this example is challenging for several main reasons,

- **High Uncertainty:** The question involves multiple unknown variables that could point to many different entities. For example, the 2021 article **U2** could point to any article published in 2021 and could only be determined given the condition **C2** and the alvei species **U1**.
- **Requirement for Exact Information Extraction:** To find the answer, the agent should list all animals mentioned on the webpages and making cross-document comparison. This would require the agent to precisely extract key information from the vast, noisy web contents, instead of simply summarizing the webpages.
- **Misleading Answers:** During the process of solving this task, there could be multiple misleading answers, such as "pigs". The agent should rigorously confirm its conclusions by checking the intended answer in all related webpages and documents.

**Existing Online RL Approaches Fail to Learn Complex Search Strategies.**   In Fig. 8, Search-R1-32B is not able to decompose the complex query into individual components, consequently only making redundant queries that involve too many unknown information. The agent also has severe hallucinations, producing conclusions that are not supported by the search results. Finally, it fails to resolve all unknown variables. This case study shows that existing online RL approaches only incentivize elementary search strategies. It is also worth noting that, since the turn limit is set as a small value, e.g. 4, during training, the model only exhibits a short tool-use horizon.

**Prompt-based LLM Agents Could Fail Due to Insufficient Capability of the LLM.**   In Fig. 8, Search-o1 (QwQ) can find the species name **U1**, as well as the 2021 article **U2** and papers **U3.1**&**U3.2** through a large amount of tool calls. However, when trying to find the answer, Search-o1 (QwQ) would easily miss key information. Consequently, the agent makes incorrect conclusions. Notably, even when the agent finds information that directly links to the correct answer, it is still misguided by previous incorrect conclusions. Finally, the agent is unable to verify the correctness of previous conclusions. This case study reveals that, though an open-source model that is not explicitly trained on agentic tasks can perform extensive tool calls, it could not make expert-level reasoning based on the retrieved contents and history contexts.

*ASearcher*-**Web-QwQ.**   We also analyze the search strategy of our end-to-end RL agent, *ASearcher*-Web-QwQ.As shown in Fig. 8, *ASearcher*-Web-QwQ decomposes the complex query into precise and focused queries. Unlike Search-o1 (QwQ) that visits a large amount of websites after each search

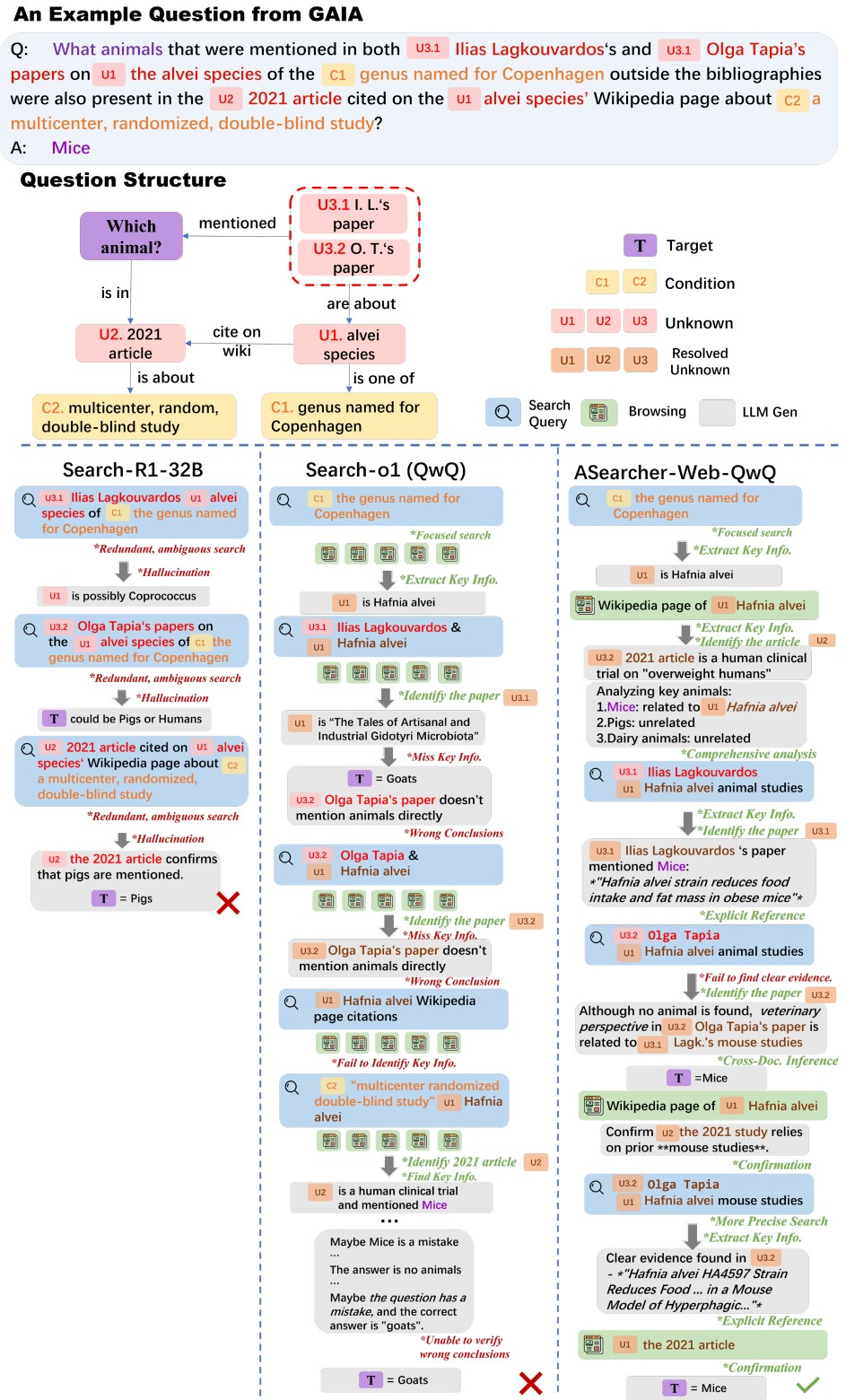

Figure 8: A case study on a complex query from GAIA. **Search-R1-32B** is unable to break down the complex question and has severe hallucinations. **Search-o1 (QwQ)** can identify the corrects articles through extensive tool calls, but easily misses key information and fails to verify wrong conclusions. Our end-to-end RL agent, *ASearcher*-**Web-QwQ**, exhibits key behaviors featuring Search Intelligence: *uncertainty-aware reasoning* (list and examine candidate answers), *precise extraction* from noisy contents, *cross-document inference*, and *rigorous confirmation*.

query, *ASearcher*-Web-QwQ focuses on visiting the most relevant website. *ASearcher*-Web-QwQ summarizes all related information from a website. Specifically, all candidate answers are listed and carefully analyzed by the agent. When trying to search for related facts in the papers **U3.1&U3.2**, the agent explicitly references the key information. When the search results do not directly point to the desired target, e.g. when searching with "Olga Tapia (**U3.2**) Hafnia alvei (**U1**) animal studies" to find the animals related to Olga Tapia's paper, the agent does not get a clear information but is able to infer the correct answer by make connection with the other paper **U3.1**. After the correct answer "Mice" is found, the agent spends further turns on confirming previous conclusions before reporting the final answer. In summary, *ASearcher* successfully train a search agent that exhibits complex behaviors that feature Search Intelligence,

- **Uncertainty-aware reasoning:** the agent exhaustively lists and examines all possibilities for uncertain entities
- **Price Key Information Extraction:** the agent is able to identify the key information from vast, noisy web contents.
- **Cross-document Inference:** the agent is able to infer critical conclusions by making connections among multiple documents.
- **Rigorous Confirmation:** the agent verifies the correctness of previous conclusions with additional tool calls.

## C  DATA SYNTHESIS AGENT

### C.1  DETAILS OF DATA SYNTHESIS

We develop a data synthesis agent to create high-quality question-answer pairs. As shown in Fig. 3, the data synthesis agent begins with a seed question, and iteratively modifies the question to increase the complexity. To ensure the synthetic question is strictly aligned with reliable sources, a list of *supporting facts* obtained during the question synthesis process is kept and continuously updated for quality verification. At each step, given the current question and a list of supporting facts, the agent automatically selects between two key actions,

- **Action 1: Injection** aims to enrich the context of the question by inserting facts related to the question. The agent first selects an entity in the question and then obtains one piece of related fact about the selected entity from external sources such as Wikipedia. Then a new question is proposed by *injecting* the fact into the question. This injection action increases complexity of the question.
- **Action 2: Fuzzing** blurs certain details in the question to increase the uncertainty level of the question. For example, "Catskill Mountain Railroad" could be replaced with "a historic mountain railway". Through fuzzing the question multiple times, both the uncertainty level and difficulty of the question would gradually increase.

To ensure that a synthetic question is of high quality and to precisely evaluate the difficulty, we incorporate a rigorous *quality verification* phase for assessing synthetic questions,

- **Step 1. Basic Quality.** We employ an LLM to assess the basic quality of each question. This verification includes checking the clarity of the question and verifying whether the question-answer pair is accurate based on the supporting facts. This quality control step ensures that each question-answer pair is properly grounded in reliable sources.
- **Step 2. Difficulty Measurement.** We employ a cutting-edge LRM (e.g., QwQ-32B) to generate multiple answers directly for the synthetic question, without using any external tool. This verification process also serves as a measure of question difficulty.
- **Step 3. Answer Uniqueness.** The fuzzing action may loosen constraints excessively, compromising the uniqueness of the answer. To prevent ambiguity resulting from multiple correct answers, we evaluate whether any of the mismatched answers generated during the Difficulty Measurement step could serve as alternative valid answers.

Through iterative injection and fuzzing, the data synthesis agent produces questions that involve complex information and high uncertainty, requiring extensive search and reasoning to find the correct answer. After completing the question synthesis process, we filter out questions that the LRM can directly generate the correct answer without relying on search tools. Since these questions can be answered solely based on the intrinsic knowledge of the model, they provide little value for enhancing search capabilities.

Starting with 14,107 seed questions, we perform an average of 6.3 injections and 3.2 fuzzes per question. From the synthetic pool, we select up to three high-quality variations per seed question. This curation process produces a final dataset of 25,624 entries, with the selected questions averaging 4.27 injections and 2.10 fuzzes each.

## C.2    EXAMPLE OF SYNTHETIC QA

Table 3: Examples of the synthetic questions, where red indicates injected facts and cyan represents fuzzed content.

| Round | Action | Question |
|---|---|---|
| **Seed QA** | - | When was Michael P. Hein born? |
| **Round 1** | Injection | When was the Eckerd College alumnus who served as the first County Executive of Ulster County, New York, and graduated with a Bachelor of Arts in Business Administration born? |
| **Round 2** | Injection | When was the individual born who, as County Executive of Ulster County, New York, permitted the Catskill Mountain Railroad to continue operations between Kingston and Hurley during the 2016 United States House of Representatives elections and also held that position during the 2018 elections? |
| **Round 3** | Fuzzing | When was the individual born who, as County Executive of Ulster County, New York, permitted a historic mountain railway to continue operations between Kingston and Hurley during the 2016 United States House of Representatives elections and also held that position during the 2018 elections? |
| ... | ... | ... |
| **Seed QA** | - | Where is the Riggs-Hamilton American Legion Post No. 20 located? |
| **Round 1** | Injection | Where is the American Legion Post in Russellville, Arkansas, built in 1934 and recognized as a notable example of WPA Rustic architecture and listed on the National Register of Historic Places located? |
| **Round 2** | Fuzzing | Where is the American Legion Post in Russellville, Arkansas, built in the early 1930s and recognized as a notable example of New Deal-era public works architecture and listed on the National Register of Historic Places located? |
| **Round 3** | Fuzzing | Where is the veterans' organization's building in Russellville, Arkansas, built in the early 1930s and recognized as a notable example of New Deal-era public works architecture and listed on the National Register of Historic Places located? |
| ... | ... | ... |

We provide two illustrative examples in Tab. 3. Starting with a simple question, the injection action replaces specific entities with related factual details. For instance, "Michael P. Hein" is expanded to "who served as the first County Executive of Ulster County, New York...". The fuzzing action introduces ambiguity by generalizing precise information, replacing the exact year "1934" with "the early 1930s" or substituting "Catskill Mountain Railroad" with "a historic mountain railway."

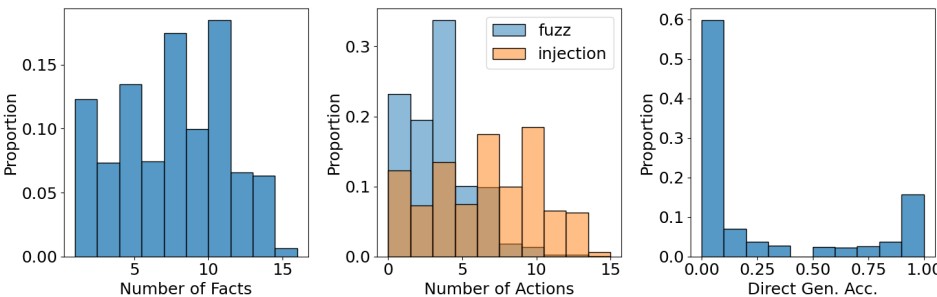

Figure 9: Statistics from our data synthesis process. (Left) The distribution of the number of supporting facts. (Middle) The distribution of the number of fuzz actions and injection actions. (Right) The accuracy distribution of QwQ-32B in answering the generated questions without using any tools.

### C.3  SYNTEHTIC QA STATISTICS

## D  PROMPTS USED FOR DATA SYNTHESIS, JUDGE, AND TEST-TIME SEARCH

---

**Prompt for Synthesis Agent Action Selection**

You are an autonomous agent for constructing general-domain QAs. Now given the current QA and relevant information. Choose one of the following action to make the question more challenging.
The current question-answer pair:
{question}
You can choose one action from the following types:
{actions}

---

**Prompt Description of SELECT (the First Step of Injection)**

SELECT: select one entity from the relevant entity list. Once such an entity is selected, an external tool will improve the difficulty of the question by replacing information about this entity in the question with sub-questions that take this entity as the answer.

If you choose SELECT, the output should be in json format:
```json
{
"action": "SELECT",
"target": url of the selected entity. note that you should only select the entity from the relevant entity list of the question and make sure the url exactly match the url in the relevant entity list.
"note": a short description of the rationale behind the selection
} ```

---

**Prompt Description of FUZZ (the Fuzzing Operation)**

FUZZ: fuzz 1 places of information in the question to make the question more challenging. Note that if you choose FUZZ, you should make sure the resulted question is still clear and has an unique answer as the original one. You should choose FUZZ only when you find certain pieces of information could clearly point to the correct answer without extensive research to find relevant information.

If you choose FUZZ, the output should be in json format:
```json
{
"action": "FUZZ",
"question": the modified question after the FUZZ operation
"note": a short description of why and how the FUZZ operation happens
}
```

---

**Prompt of Combining Two Questions (the Second Step of Injection)**

You are an autonomous agent for constructing general-domain QAs. Now given two questions, combine these questions into one. Specially, the answer to the second question is related to some entity in the first question. To make the combined question challenging, you need to remove information about the answer of the second question and ensure the answer of the combined question remains the same as the answer of the first question. Please ensure that the combined question is clear, solvable, and has an unique answer.

The first Question: ```
{questionA}
```

The second Question:
```
{questionB}
```

Relevant statements:
```
{statements}
```

The output should be in json format:
```json
{
"question": the proposed question,
"answer": the answer
"note": one short description of how the two questions are combined
}
```

---

**Prompt of Basic Quality Check**

Check the validity of this question-answer pair given its relevant information.

The question is valid is and only if:
1. the question is not a simple concatenation of two or more questions
2. the provided answer is the only correct answer to the question
3. the question has an unique answer
4. the question can be solved based on the relevant statements

The question-answer pair and the relevant information:
{question}

You should reply "yes" or "no" indicating the validity of the question-answer pair. You should think step-by-step first before the final judgement in json format:

Analysis

// your analysis

Final Judgement
```json
{
"judgement": "yes" or "no"
}
```

---

**Prompt of Checking Whether an Answer Could be an Alternative Answer**

Determine whether the predicted answer is also correct to the question. Specially, the predicted answer is different from the ground-truth answer, and you should check whether the predicted answer fits with all constraints in the question and is also a correct answer to the question.

Question: {question}

Ground-truth answer: {gt_answer}

Facts supporting the ground-truth answer:
```txt
{statements}
```

Predicted answer: {pred_answer}

```json
{
"judgement": "yes" or "no"
}
```

---

---

**Prompt of LLM-as-Judge**

You are an evaluation assistant. Please determine if the predicted answer is equivalent to the labeled answer.

Question: {question}

Labeled Answer: {gt_answer}

Predicted Answer: {pred_answer}

Did the model give an answer **equivalent** to the labeled answer? Please respond with "Correct" if they are equivalent, or "Incorrect" if they are not equivalent.

The output should in the following json format:
```json
{
"rationale": your rationale for the judgement, as a text,
"judgement": your judgement, can only be "Correct" or "Incorrect",
}
```

---

# E    ADDITIONAL RESULTS

**Applying *ASearcher* Training on a Small Non-Reasoning Model.**    We apply our training recipe on **Qwen2.5-7B-Instruct**, a non-reasoning model with much smaller scale. The results show that our recipe is also able to enhance the search capability of a small, non-reasoning model.

Table 4: Results on Qwen2.5-7B-Instruct

|            | GAIA        | xBench      |
|------------|-------------|-------------|
| Before RL  | 18.2        | 23.0        |
| After RL   | 27.7 (+9.5) | 27.8 (+4.8) |

This demonstrates that,

- Our pipeline works effectively on smaller models. Pure RL training also works for non-reasoning models.
- The significant performance improvement is not solely due to the choice of the base model, QwQ, but comes from the full training recipe.

---

**Prompt for Aggregating $K$ Independently Generated Trajectories in Test-time Search**

Given a question and $K$ independently generated results, please determine the most reliable answer.

Question:
{question}

Result 1:
{result_1}

Result 2:
{result_2}

. . .

Result K:
{result_K}

The output should in the following json format:
"'json
{
"rationale": your rationale as a text,
"answer": the most reliable final answer,
}

---

**Evaluating Baseline with Summary Tool & Test-time Scaling.** We conduct a comparison with WebSailor-32B (Li et al., 2025a) when using DeepSeek-V3 for summary and test-time search with $K = 16$, under a turn limit of 128. Since WebSailor does not produce a comprehensive analysis at the final step, we directly feed all the trajectories into DeepSeek-V3 to determine the most reliable answer when performing Test-time Search. The results show that:

- The baseline, WebSailor-32B, only conducts short-horizon search, and does not show an advanced capability of utilizing more tool calls for rigorous conclusion verification.

- When employing test-time augmentation approaches, ASearcher significantly outperforms WebSailor-32B. We hypothesize that this is because WebSailor does not produce a compact analysis, and, therefore, the most reliable answer could not be easily determined from a set of noisy trajectories.

| Method | GAIA | xBench |
|---|---|---|
| ASearcher (Ours) | 58.7 | 51.1 |
| + Summary w. DeepSeek-V3 & Test-time Search ($K = 16$) | **71.8** | **75.0** |
| WebSailor-32B | 53.2 | 53.3 |
| + Summary w. DeepSeek-V3 & Test-time Search ($K = 16$) | 60.2 | 64.0 |

Table 5: WebSailor-32B with Test-Time Augmentations

| Method | GAIA | xBench |
|---|---|---|
| ASearcher (Ours) | **13.81** | **13.67** |
| WebSailor-32B | 4.24 | 5.17 |

Table 6: Number of Tool Calls of ASearcher & WebSailor-32B

