# OpenReview forum: "Unlocking Long-Horizon Agentic Search with Large-Scale End-to-End RL"
_ICLR.cc/2026/Conference — ICLR 2026 Poster_

### Official Review · Reviewer_JUpp · 2025-10-19

**Soundness:** 2
**Presentation:** 3
**Contribution:** 2
**Rating:** 2
**Confidence:** 4

**Summary:**

This paper introduces ASearch, a single-model search agent purely trained by RL without using commercial APIs for data or tools. Firstly, a data synthesis agent is used to apply question injection and fuzzing to open-source datasets, increasing the complexity of training data, followed by three steps of quality verification. Secondly, a two-stage curriculum RL training is employed. The paper further accelerates the training process by asynchronizing both trajectory rollout and training. Experiments show that ASearch can achieve better performance compared to many strong baselines by using only an open-source 32B model.

**Strengths:**

ASearch can achieve excellent results without relying on API model-synthesized trajectories, and the training process is fully RL without cold-start. The proposed asynchronous rollout and training strategy can effectively enhance training efficiency, enabling the model to make more tool calls.

**Weaknesses:**

1. There are several crucial details in the method that are unclear. For example: 1) How is the uniqueness of answers determined during the quality verification process? How does the model know if an answer is unique without executing an action? 2) How are the answers for the synthesized new questions determined? 3) How is question injection and fuzzing implemented, and what prompts are used?

2. The scalability of the method in this paper is questionable. Although the paper does not rely on commercial API models, QwQ-32B already possesses sufficient reasoning capabilities compared with most open-source models. If replaced with models at the 7B or 14B level, can the pipeline in this paper successfully boost the model's performance? Furthermore, the data synthesis in this paper relies on existing open-source datasets. The community is more concerned about how to construct a large number of high-quality and complex query-answer pairs without relying on open-source data.

3. There are risks regarding the fairness of the baseline comparisons in Table 1. For example, do the various baselines use the same training data? Additionally, this paper allows a maximum of 128 tool calls, far more than other baselines. Does this setting itself bring significant performance gains?

4. Furthermore, the significant performance improvement achieved through zero RL in this paper may be related to QwQ being a reasoning model. Would switching to a non-reasoning model imply the need for a cold start?

5. Most of the analytical experiments in this paper are case studies and lack strong persuasiveness.

**Questions:**

1. Many prompts used in this paper have not been disclosed, such as the prompts for question injection and fuzzing, quality verification, LLM-as-judge, and others.

2. How would it impact if all baselines were allowed a maximum of 128 rounds of tool calls?

3. How would the effects be if other baselines used Summary or TTS? How is TTS specifically implemented - based on self-consistency or pass@K?

4. The reasons behind this paper's ability to surpass baselines are not very clear. Why can using a 32B model to synthesize QA and training with zero-RL achieve better results than other baselines using stronger models to synthesize trajectories with a cold start + RL approach?

---

> ### Author Response · Authors · 2025-11-26
> **Response to Reviewer JUpp (Part I)**
>
> Thank you for your detailed feedback. We address the core concerns regarding implementation details, scalability, and ablations by providing new experimental results and clarifications.
>
> ### W1: Missing implementation details
>
> We have added Appendix C with complete implementation details of the data synthesis process, and Appendix D with prompts for question injection, fuzzing, quality verification, and LLM-as-judge. Additionally, we have also provided all our code for training, evaluation, and question synthesis, as well as the training data, in [this anonymous link](https://anonymous.4open.science/r/ASearcher-7022) and supplementary materials.
>
> #### (1) How is the uniqueness of answers determined during quality verification?
>
> Answer uniqueness is verified through applying the LLM to judge whether there exists any answer generated during the difficulty measurement step is also correct to the input QA, based on the latest version of the question and related facts. This step aims to prevent fuzzing steps from creating vague questions that have multiple valid answers.
>
> **An Example of Fuzzing Creating a Vague Question:**
> - *Question before fuzzing* could be "In year 1999, what major event related to `PersonA` happened?". There are two related facts, "In year 1998, `EventA` happened." and "In year 1999, `EventB` happened." The answer should be `EventB`.
> - *Question after fuzzing* may become "**During year 1990-2000**, what major event related to `PersonA` happened?" Based on the related facts, the LLM can conclude that both `EventA` and `EventB` are correct for the new question. This question is rejected during synthesis.
>
> #### (2) How are the answers for the synthesized new questions determined?
>
> The synthesis process starts from a seed question-answer pair. During the synthesis process, **the answer is kept unchanged** while only the question is modified with increased complexity and difficulty. In the "Step 1. Basic Quality Check" step, the LLM evaluates whether the answer is correct to the question based on related facts to ensure that the synthesized question and the answer are aligned.
>
> #### (3) How is question injection and fuzzing implemented, and what prompts are used?
>
> We have added detailed prompts for question injection, fuzzing, quality verification, and LLM-as-judge in Appendix D. The code of data synthesis can also be found in [this anonymous link](https://anonymous.4open.science/r/ASearcher-7022) and supplementary materials.
>
> ### W2: Scalability to smaller models and data synthesis without open-source datasets
>
> #### (1) Scalability to 7B model
>
> We apply our training recipe on Qwen2.5-7B-Instruct, a non-reasoning model with much smaller scale.
>
> **Table A: Results on Qwen2.5-7B-Instruct**
>
> |  | GAIA | xBench |
> |------------|------|--------------|
> | Before RL | 18.2 | 23.0 |
> | After RL | 27.7 (+9.5) | 27.8 (+4.8) |
>
> The experiement result demonstrates that:
> - Our pipeline works effectively on smaller models. **Pure RL training also works for non-reasoning models**.
> - The significant performance improvement is not solely due to the choice of the base model, QwQ, but comes from the full training recipe.
>
> #### (2) Data synthesis without open-source datasets
>
> We acknowledge this is an important direction. Our work focuses on showing a training recipe that can incentivize competitive search capability without commercial LLMs. While we currently use Wikipedia 2018 as the knowledge source, our **synthesis pipeline is general and can be applied to other knowledge sources**, such as priopertory knowledge bases, in-domain data, and dynamic updated web contents. We view automatic dataset construction from diverse sources as valuable future work.

---

> > ### Author Response · Authors · 2025-11-26
> > **Response to Reviewer JUpp (Part II)**
> >
> > ### W3: Fairness of baseline comparisons
> >
> > We address each fairness concern:
> >
> > #### (1) Do baselines use the same training data?
> >
> > The results of baseline methods are directly obtained from their official reports, which is a common practice for reporting baseline performance in prior works [1,2,3]. Each baseline is trained with its own specific training data. Since the training code and data of competitive baselines like WebSailor [1] and WebShaper [3] are not open-sourced, retraining them on our data is infeasible.
> >
> > #### (2) Does the 128 tool calls setting bring significant performance gains?
> >
> > We have added ablation studies on the training-time turn limit to understand its impact.
> >
> > **Table B: Ablation Study on Turn Limit**
> >
> > |      | # of Tool Calls at Training Time | GAIA | xBench |
> > |---------------|------|------|--------|
> > | **Full System** | | | |
> > | ASearcher (Full) w. Turn Limit=128 | 26.59 | 58.7 | 51.1 |
> > | | | | |
> > | **Ablating Training-time Turn Limit** | | | |
> > | ASearcher w. Turn Limit=10 | 3.48 | 49.2 | 39.3 |
> >
> > Results in Table B highlight the necessity of a large turn limit at training time. This provides sufficient exploration space for the agent to learn advanced, long-horizon search strategies that involve increased tool calls for rigorous verification, as detailed in our case study in Sec 4.3.
> >
> > ### W5: Analytical experiments beyond case studies
> >
> > In addition to case studies, we also conduct comprehensive ablation studies (see Table C) to systematically evaluate the contribution of different components. To ensure ASearcher and all ablations use the same amount of training steps, the ablation study is taken by starting from the Stage 1 checkpoint of ASearcher at the 200th training step, with additional 200-step training, .
> >
> > We highlight key conclusions from our ablation study,
> > - **Training-time Turn Limit.** A large turn limit is the key to enable the long-horizon search capability of the model. Training with a large turn limit significantly surpasses training with a small turn limit.
> > - **Two-Stage Curriculum.** Compared with continued training with only Stage 1 data, the full training recipe with a two-stage curriculum helps the agent to learn stable long-horizon search by focusing on challenging queries that require at least 5 tool calls.
> > - **Data Quality.** Finally, we also compare our training data with the baseline, AFM[2], by applying asynchrounus RL training using a 128 turn limit with training data from AFM. Training with AFM data is unable to incentivize complex search capabilities and achieves sub-optimal benchmark performance, showing that our RL data is of high quality.
> >
> > **Table C: Ablation Study (Complete)**
> >
> > |  | # of Tool Calls at Training Time| GAIA | xBench |
> > |---------------|---|------|--------|
> > | **Full System** | | | |
> > | ASearcher (Full) | 26.59 | 58.7 | 51.1 |
> > | | | | |
> > | **Ablating Training-time Turn Limit** | | | |
> > | ASearcher w. Turn Limit=10 | 3.48 | 49.2 | 39.3 |
> > | | | | |
> > | **Ablating Training Data** |  | | | |
> > | ASearcher w. Stage 1 Data Only | 5.40 | 51.6 | 43.0 |
> > | ASearcher w. AFM Data | 4.12 | 50.9 | 39.9 |
> >
> >
> > ### Q3: Results of Baseline Using DeepSeek-V3 Summary & TTS
> >
> > **Implementation Details of TTS:** TTS is implemented by first generating $K$ independent trajectories. An external LLM (DeepSeek-V3) is then queried to determine the most reliable answer given the last-step analysis of all trajectories.
> >
> > **Evaluating WebSailor-32B with Test-time Augmentations:** We conduct a comparison with WebSailor-32B [1] when using DeepSeek-V3 for summary and test-time search with K=16, under a turn limit of 128. Since WebSailor does not produce a comprehensive analysis, we directly feed all the trajectories into DeepSeek-V3 for the final judgment. The results show that:
> > - The baseline, WebSailor-32B, only conducts short-horizon search, and does not show an advanced capability of utilizing more tool calls for conclusion verification.
> > - When employing test-time augmentation approaches, ASearcher significantly outperforms WebSailor-32B. We hypothesize that this is because WebSailor does not produce a compact analysis and therefore the most reliable answer could not be easily determined from a set of noisy trajectories.
> >
> >
> > **Table D.1: WebSailor-32B with Test-Time Augmentations**
> >
> > |       | GAIA | xBench |
> > |---------------|------|------|
> > | ASearcher (Ours) | 58.7 | 51.1 |
> > | + Summary w. DeepSeek-V3 & Test-time Search (K=16) | 71.8 | 75.0 |
> > | | | |
> > | WebSailor-32B | 53.2 | 53.3 |
> > | + Summary w. DeepSeek-V3 & Test-time Search (K=16) | 60.2 | 64.0 |
> >
> > **Table D.2: Number of Tool Calls of ASearcher & WebSailor-32B**
> >
> > |       | GAIA | xBench |
> > |---------------|------|------|
> > | ASearcher (Ours) | 13.81 | 13.67 |
> > | WebSailor-32B | 4.24 | 5.17 |

---

> > > ### Author Response · Authors · 2025-11-26
> > > **Response to Reviewer JUpp (Part III)**
> > >
> > > ### Q4: Why can ASearcher surpass baselines using stronger commercial models?
> > >
> > > We highlight the key reasons of the success of ASearcher:
> > > 1. **RL discovers advanced long-horizon search strategies:** RL training enables the model to learn to use multiple tool calls for **rigorous analysis and verifications of previous conclusions**, as shown in our case study in Sec 4.3. This capability is directly supported by the data in Table D.2, where ASearcher uses $\sim 3\times$ tool calls as WebSailor-32B.
> > > 2. **High-quality Training Data:** As shown in our additional ablation study on the training data, our training data outperforms baseline methods such as AFM, allowing the emergence of complex search strategies, as shown in Table C.
> > > 3. **Large Training-time Turn Limit:** Besides data quality, a turn limit much larger than prior works, e.g. Search-R1 uses a turn limit of less than 10, provides a sufficient exploration space for the agent to learn to conduct long-horizon search, as shown in Table B.
> > >
> > > ## Summary
> > >
> > > We have significantly strengthened the paper with:
> > >
> > > 1. **Complete prompt details** (Appendix D): All prompts used by ASearcher.
> > > 2. **Scalability experiments** (Appendix E): Results on Qwen2.5-7B-Instruct showing pipeline effectiveness on smaller, non-reasoning models
> > > 3. **Comprehensive ablation studies** (Sec 4.4): Demonstrating contribution of training-time turn limit and training data
> > > 4. **Baseline comparisons** (Appendix E): Including comparison with WebSailor-32B with test-time augmentations
> > >
> > > We emphasize that our primary contribution is showing that our training recipe can incentivize competitive search capability of an open-source model without relying on superior commercial models. Our work reveals that RL training enables the emergence of complex long-horizon search strategies, such as rigorous analysis and verification, as evidenced by significant increasement of the amount of tool calls through training and dedicated case studies. Notably, **our finding on search agents is akin to DeepSeek-R1 [4], where the emergent reasoning capabilities can be fully incentivized by RL.**
> > >
> > > We hope these additions address your concerns and would be happy to provide any additional clarifications.
> > >
> > > [1] Li, K., Zhang, Z., Yin, H., Zhang, L., Ou, L., Wu, J., ... & Zhou, J. (2025). WebSailor: Navigating Super-human Reasoning for Web Agent. arXiv preprint arXiv:2507.02592.
> > >
> > > [2] Li, W., Lin, J., Jiang, Z., Cao, J., Liu, X., Zhang, J., ... & Zhou, W. (2025). Chain-of-agents: End-to-end agent foundation models via multi-agent distillation and agentic rl. arXiv preprint arXiv:2508.13167.
> > >
> > > [3] Li, X., Jin, J., Dong, G., Qian, H., Wu, Y., Wen, J. R., ... & Dou, Z. (2025). Webthinker: Empowering large reasoning models with deep research capability. arXiv preprint arXiv:2504.21776.
> > >
> > > [4] Guo, D., Yang, D., Zhang, H., Song, J., Wang, P., Zhu, Q., ... & Tan, Y. (2025). Deepseek-r1 incentivizes reasoning in llms through reinforcement learning. Nature, 645(8081), 633-638.

---

> ### Author Response · Authors · 2025-11-28
>
> Dear Reviewer,
>
> I hope this message finds you well. As the discussion period is nearing its end, with **less than one week remaining**. I wanted to ensure that we have addressed your concerns satisfactorily. If there are any additional points or feedback you'd like us to consider, please let us know. Your insights are invaluable to us, and we're eager to address any remaining issues to improve our work.
>
> Thank you for your time and effort in reviewing our paper.

---

### Official Review · Reviewer_yr7v · 2025-10-31

**Soundness:** 3
**Presentation:** 3
**Contribution:** 3
**Rating:** 6
**Confidence:** 3

**Summary:**

In this paper, the authors propose an LLM-based search agent named ASearcher, which is built from an open-source model (QwQ-32B). It is trained purely using end-to-end Reinforcement Learning (RL). Unlike many existing approaches, this work avoids any reliance on commercial LLM APIs for generating supervised fine-tuning data or for serving as specialized sub-modules. The paper provides a novel data synthesis agent along with a large-scale, long-horizon RL training framework. The proposed ASearcher agent achieves state-of-the-art performance among 32B-scale open-source agents on benchmarks like GAIA, xBench, and Frames.

**Strengths:**

1. In general, the paper is well written with good structure and is easy to follow.

2. In this paper, the authors are trying to tackle an important and challenging task. Training agents that perform deep, long-horizon web search is timely and practically useful. The paper frames these challenges very well and clearly.

3. The proposed method demonstrates that a single open-model (QwQ-32B) trained purely with RL, which has no commercial LLMs for data or submodules, can learn sophisticated multi-turn search strategies is a meaningful contribution to the RL-for-agents literature. The result challenges the current trend of heavily relying on multi-model/commercial LLM pipelines.

4. The injection plus fuzzing loop, combined with verification checks, is a practical method to produce hard, tool-use-requiring QA pairs. The authors also provide useful statistics and examples in the appendix.

**Weaknesses:**

1. It would be better if more ablation studies could be provided. The paper presents two major contributions: the novel QA synthesis pipeline and the long-horizon asynchronous RL system. However, their relative importance is not disentangled.

2. The data synthesis agent uses external sources to support facts and then rejects questions solvable without tools. But more detail is expected: what external sources were used (Wikipedia only?), how is answer uniqueness verified, and what mechanisms prevent the synthetic QA from leaking evaluation set contents or being too close to benchmark questions?

3. The proposed system relies on a very large language model (QwQ-32B) and extensive GPU hours (16k H800 GPU-hours), which is far more expensive than simply using commercial APIs for data generation or inference.

4. I wonder if the proposed method can perform well on the other common base models.

**Questions:**

Please check the issues and questions mentioned in the "Weaknesses" section.

---

> ### Author Response · Authors · 2025-11-26
> **Response to Reviewer yr7v**
>
> Thank you for your detailed feedback. We address your concerns below:
>
> ### W1: Ablation Study
>
> In addition to case studies, we also conduct comprehensive ablation studies (see Table A) to systematically evaluate the contribution of different components. The ablation study starts from the Stage 1 checkpoint at the 200th training step, followed by an additional 200 steps of training under different settings to ensure a fair comparison.
>
> We highlight key conclusions from our ablation study,
> - **Training-time Turn Limit.** Training with a large turn limit significantly surpasses training with a small turn limit. This shows that the RL agent must be given the training budget (in terms of turns) to discover and reinforce long-horizon search strategies.
> - **Two-Stage Curriculum.** Compared with continued training with only Stage 1 data, the full training recipe with a two-stage curriculum helps the agent to learn stable and effective strategies by focusing on challenging queries that require at least 5 tool calls.
> - **Data Quality.** Finally, we also compare our training data with the baseline, AFM[1], by applying asynchrounus RL training using a 128 turn limit with training data from AFM. Training with AFM data is unable to incentivize complex search capabilities and achieves sub-optimal benchmark performance, showing that our RL data is of high quality.
>
> **Table A: Ablation Study**
>
> |  | # of Tool Calls at Training Time| GAIA | xBench |
> |---------------|---|------|--------|
> | **Full System** | | | |
> | ASearcher (Full) | 26.59 | 58.7 | 51.1 |
> | | | | |
> | **Ablating Training-time Turn Limit** | | | |
> | ASearcher w. Turn Limit=10 | 3.48 | 49.2 | 39.3 |
> | | | | |
> | **Ablating Training Data** |  | | | |
> | ASearcher w. Stage 1 Data Only | 5.40 | 51.6 | 43.0 |
> | ASearcher w. AFM Data | 4.12 | 50.9 | 39.9 |
>
> ### W2: Data synthesis details
>
> We have added Appendix C with complete details.
>
> #### External Sources Used
>
> We use the Wikipedia 2018 dump as the data source for data synthesis.
>
> #### Answer Uniqueness Verification
>
> Answer uniqueness is verified through applying the LLM to judge whether there exists any answer generated during the difficulty measurement step is also correct to the input QA, based on the latest version of the question and related facts. This step aims to prevent fuzzing steps from creating vague questions that have multiple valid answers.
>
> **An Example of Fuzzing Creating a Vague Question:**
> - *Question before fuzzing* could be "In year 1999, what major event related to `PersonA` happened?". There are two related facts, "In year 1998, `EventA` happened." and "In year 1999, `EventB` happened." The answer should be `EventB`.
> - *Question after fuzzing* may become "**During year 1990-2000**, what major event related to `PersonA` happened?" Based on the related facts, the LLM can conclude that both `EventA` and `EventB` are correct for the new question. This question is rejected during synthesis.
>
> #### Evaluation Set Leakage Prevention
>
> We do not include an explicit evaluation set leakage prevention since the data synthesis process is fully taken on a offline dataset, the Wikipedia 2018 dump, while the evaluated benchmarks such as GAIA and xBench typically require information after 2018 (i.e., information not present in the dump). This inherent temporal separation provides a robust guarantee against data leakage.
>
>
> ### W3: Computational cost concerns
>
> We understand the concern regarding the $16k$ H800 GPU hours. We emphasize that this significant effort was necessary to establish the state-of-the-art capability for an open-source model on challenging deep search tasks, proving the effectiveness of the our RL recipe.
>
> Furthermore, we demonstrate the practicality of our approach through additional experiments on a smaller model, Qwen2.5-7B-Instruct. Please refer to Table B for results on Qwen2.5-7B-Instruct.
>
> ### W4: Generalization to other models
>
> We apply our training recipe on Qwen2.5-7B-Instruct, a model with much smaller scale. The results show that our recipe is also able to enhance the search capability of a small model.
>
> **Table B: Results on Qwen2.5-7B-Instruct**
>
> |  | GAIA | xBench |
> |------------|------|--------------|
> | Before RL | 18.2 | 23.0 |
> | After RL | 27.7 (+9.5) | 27.8 (+4.8) |
>
> The results show that our full training pipeline works effectively on a smaller model, achieving a significant performance gain on both benchmarks. This validates that the performance improvement is not solely due to the base model size (QwQ-32B) but is a direct result of the RL training recipe and its ability to learn complex search policies.
>
> We believe that these clarifications and new results address your concerns and would be happy to provide any additional clarifications.
>
> [1] Li, W., Lin, J., Jiang, Z., Cao, J., Liu, X., Zhang, J., ... & Zhou, W. (2025). Chain-of-agents: End-to-end agent foundation models via multi-agent distillation and agentic rl. arXiv preprint arXiv:2508.13167.

---

> ### Author Response · Authors · 2025-11-28
>
> Dear Reviewer,
>
> I hope this message finds you well. As the discussion period is nearing its end, with **less than one week remaining**. I wanted to ensure that we have addressed your concerns satisfactorily. If there are any additional points or feedback you'd like us to consider, please let us know. Your insights are invaluable to us, and we're eager to address any remaining issues to improve our work.
>
> Thank you for your time and effort in reviewing our paper.

---

### Official Review · Reviewer_HJhQ · 2025-11-01

**Soundness:** 3
**Presentation:** 2
**Contribution:** 2
**Rating:** 4
**Confidence:** 4

**Summary:**

This paper presents ASearcher, a single-model search agent trained with large-scale reinforcement learning , demonstrating long-horizon reasoning and competitive performance.

**Strengths:**

1. This paper is generally well-organized.
2. The paper involves a substantial amount of engineering work, including large-scale RL training, data synthesis.
3. Demonstrates impressive long-horizon reasoning performance.

**Weaknesses:**

1. While the paper emphasizes end-to-end RL for long-horizon search, similar ideas have already been explored by Search-R1, WebThinker, websailor, and Chain-of-Agents, which also use RL or multi-agent training to enhance tool use. The main difference here fully asynchronous RL is more of a systems optimization than a conceptual breakthrough in agent reasoning or policy learning. While technically well-executed, the contribution is rather straightforward and does not introduce substantial new insights

2.  The reported 16k H800 GPU hours make the approach extremely resource-intensive and difficult to reproduce, especially for academic or smaller research groups, which greatly limits its practical contribution and accessibility to the broader community.

3. The paper’s contribution, scope, and novelty are quite unclear. It reads more like a well-executed technical report or open-source project than a research paper with distinct conceptual advances. The work mainly combines and scales up ideas that are already known in the community. Moreover, the performance gains might largely reflect the inherent capability of QwQ-32B and the heavy use of synthetic QA generation and data filtering, rather than the RL algorithm itself.

4. It is also unclear what training setup was used for the baselines—did they use the same data, or were they retrained? If the goal is to compare RL algorithms, the training data and environment should be kept identical; if the goal is to emphasize data synthesis or agentic setup, experiments with smaller models would help validate that claim.

5. In addition, I did not see any ablation studies analyzing the contribution of each component (e.g., data quality, RL updates), which makes it difficult to attribute where the actual improvements come from.

6. The entire pipeline feels more like a collection of separately optimized components rather than a truly “end-to-end” system

**Questions:**

Please refer to the weaknesses section

---

> ### Author Response · Authors · 2025-11-26
> **Response to Reviewer HjhQ (Part I)**
>
> We appreciate the reviewer's detailed feedback. We address the core concerns regarding novelty, computational cost, and ablations by providing new experimental results and clarifications. We respectfully argue that ASearcher offers a unique and substantial contribution by demonstrating a full, competitive **Reinforcement Learning (RL) training recipe for an open-source search agent**, which enables the **emergence of complex, long-horizon reasoning capabilities**.
>
> ### Unique Contribution and Novelty (W1, W3, W6)
>
> While acknowledging that ASearcher shares some similarity with related works in terms of end-to-end RL and data synthesis, our contribution is distinct and moves beyond a simple systems optimization.
> - **RL-Enabled Emergence of Long-Horizon Strategy**: Our work uniquely demonstrates that **Reinforcement Learning enables the emergence of complex long-horizon search strategies** in an open-source model, such as rigorous multi-step verification and nuanced analysis. **This finding is akin to the key insight of DeepSeek-R1** [1], where the emergent reasoning capabilities can be fully incentivized by RL.
> - **Full, Competitive Open-Source Recipe**: We present a full, end-to-end RL training recipe that achieves advanced performance on demanding benchmarks **without relying on commercial models**.
> - Through additional ablation studies (Sec 4.4), we identify that **a large training-time turn limit** and **high-quality synthetic data** are the key to unlock this long-horizon search capability.
>
> ### Computational Cost and Accessibility (W2)
>
> We understand the concern regarding the $16k$ H800 GPU hours. We emphasize that this significant effort was necessary to establish the state-of-the-art capability for an open-source model on challenging deep search tasks, proving the effectiveness of the our RL recipe.
>
> To address the accessibility concern, we conducted new experiments using a smaller model, Qwen2.5-7B-Instruct.
>
> **Table A: Results on Qwen2.5-7B-Instruct**
>
> |  | GAIA | xBench |
> |------------|------|--------------|
> | Before RL | 18.2 | 23.0 |
> | After RL | 27.7 (+9.5) | 27.8 (+4.8) |
>
> The results show that our full training pipeline works effectively on a smaller model, achieving a significant performance gain on both benchmarks. This validates that the performance improvement is not solely due to the base model size (QwQ-32B) but is a direct result of the RL training recipe and its ability to learn complex search policies
>
> ### Comprehensive Ablation Studies (W3, W5)
>
> We have added comprehensive ablation studies to systematically evaluate the contribution of different components, directly addressing the reviewer's request. The ablation study starts from the Stage 1 checkpoint at the 200th training step, followed by an additional 200 steps of training under different settings to ensure a fair comparison.
>
> We highlight key conclusions from our ablation study,
> - **Training-time Turn Limit.** Training with a large turn limit significantly surpasses training with a small turn limit. This shows that the RL agent must be given the training budget (in terms of turns) to discover and reinforce long-horizon search strategies.
> - **Two-Stage Curriculum.** Compared with continued training with only Stage 1 data, the full training recipe with a two-stage curriculum helps the agent to learn stable and effective strategies by focusing on challenging queries that require at least 5 tool calls.
> - **Data Quality.** Finally, we also compare our training data with the baseline, AFM[2], by applying asynchrounus RL training using a 128 turn limit with training data from AFM. Training with AFM data is unable to incentivize complex search capabilities and achieves sub-optimal benchmark performance, showing that our RL data is of high quality.
>
> **Table B: Ablation Study**
>
> |  | # of Tool Calls at Training Time| GAIA | xBench |
> |---------------|---|------|--------|
> | **Full System** | | | |
> | ASearcher (Full) | 26.59 | 58.7 | 51.1 |
> | | | | |
> | **Ablating Training-time Turn Limit** | | | |
> | ASearcher w. Turn Limit=10 | 3.48 | 49.2 | 39.3 |
> | | | | |
> | **Ablating Training Data** |  | | | |
> | ASearcher w. Stage 1 Data Only | 5.40 | 51.6 | 43.0 |
> | ASearcher w. AFM Data | 4.12 | 50.9 | 39.9 |

---

> > ### Author Response · Authors · 2025-11-26
> > **Response to Reviewer HjhQ (Part II)**
> >
> > ### W4: Setup of and Comparison with Baselines
> >
> > The results of baseline methods are directly obtained from their official reports, which is a common practice for reporting baseline performance in prior works [2,3,4]. Each baseline method are trained with its own training data. Since the training code and training data of competitive baselines such as WebSailor[3] and WebShaper[4] are not open-sourced, it is infeasible to retrain these baselines.
> >
> > Through comparison with the baselines, our goal is to demonstrate that:
> > - **RL Competitiveness**: RL can incentivize competitive search capability in an open-source model competitive with other advanced open-source agents, **without relying on commercial models**.
> > - **Emergence of Long-Horizon Policy**: ASearcher achieves its competitive performance by learning to utilize **significantly more tool calls** at test time for comprehensive verification and analysis, a key feature that is missing in the other baselines.
> >
> > **Evaluating Baseline with Test-time Augmentations:** We additionally conduct a comparison with WebSailor-32B [3] when using DeepSeek-V3 for summary and test-time search with K=16, under a turn limit of 128. Since WebSailor does not produce a comphrensive analysis at the final step, we directly feed all the trajectories into DeepSeek-V3 to determine the most reliable answer when performing Test-time Search. The results show that:
> > - The baseline, WebSailor-32B, only conducts short-horizon search, and does not show a advanced capability of utilizing more tool calls for conclusion verification.
> > - When employing test-time augmentation approaches, ASearcher significantly outperforms WebSailor-32B. We hypothesize that this is because WebSailor does not produce a compact analysis and therefore the most reliable answer could not be easily determined from a set of noisy trajectories.
> >
> >
> > **Table C.1: WebSailor-32B with Test-Time Augmentations**
> >
> > |       | GAIA | xBench |
> > |---------------|------|------|
> > | ASearcher (Ours) | 58.7 | 51.1 |
> > | + Summary w. DeepSeek-V3 & Test-time Search (K=16) | 71.8 | 75.0 |
> > | | | |
> > | WebSailor-32B | 53.2 | 53.3 |
> > | + Summary w. DeepSeek-V3 & Test-time Search (K=16) | 60.2 | 64.0 |
> >
> > **Table C.2: Number of Tool Calls of ASearcher & WebSailor-32B**
> >
> > |       | GAIA | xBench |
> > |---------------|------|------|
> > | ASearcher (Ours) | 13.81 | 13.67 |
> > | WebSailor-32B | 4.24 | 5.17 |
> >
> > ### W6: End-to-End Reinforcement Learning
> >
> > Although the full training recipe contains multiple components, we have not claimed our work presents an end-to-end system. Instead, we emphasize that the search agent is trained with reinforcement learning in an end-to-end manner. Specifically, **all reasoning texts, analysis, and actions** generated by the model is optimized through the RL objective, which is computed based on the correctness of the final answer.
> >
> >
> > **Conclusion**: Our work is not a simple combination of existing components, but a substantial engineering and experimental effort that proves a full RL training recipe can unlock complex, long-horizon search capabilities in open-source models. The new ablation studies, smaller-model experiment, and detailed comparison on tool-call usage address the reviewer's concerns and highlight the conceptual novelty and effectiveness of our training recipe.
> >
> > We believe that these clarifications and new results address your concerns and would be happy to provide any additional clarifications.
> >
> > [1] Guo, D., Yang, D., Zhang, H., Song, J., Wang, P., Zhu, Q., ... & Tan, Y. (2025). Deepseek-r1 incentivizes reasoning in llms through reinforcement learning. Nature, 645(8081), 633-638.
> >
> > [2] Li, W., Lin, J., Jiang, Z., Cao, J., Liu, X., Zhang, J., ... & Zhou, W. (2025). Chain-of-agents: End-to-end agent foundation models via multi-agent distillation and agentic rl. arXiv preprint arXiv:2508.13167.
> >
> > [3] Li, K., Zhang, Z., Yin, H., Zhang, L., Ou, L., Wu, J., ... & Zhou, J. (2025). WebSailor: Navigating Super-human Reasoning for Web Agent. arXiv preprint arXiv:2507.02592.
> >
> > [4] Li, X., Jin, J., Dong, G., Qian, H., Wu, Y., Wen, J. R., ... & Dou, Z. (2025). Webthinker: Empowering large reasoning models with deep research capability. arXiv preprint arXiv:2504.21776.

---

> ### Author Response · Authors · 2025-11-28
>
> Dear Reviewer,
>
> I hope this message finds you well. As the discussion period is nearing its end, with **less than one week remaining**. I wanted to ensure that we have addressed your concerns satisfactorily. If there are any additional points or feedback you'd like us to consider, please let us know. Your insights are invaluable to us, and we're eager to address any remaining issues to improve our work.
>
> Thank you for your time and effort in reviewing our paper.

---

### Official Review · Reviewer_1mWo · 2025-11-11

**Soundness:** 3
**Presentation:** 4
**Contribution:** 3
**Rating:** 6
**Confidence:** 4

**Summary:**

This work proposed a strong training method where the performance gain comes mostly from an improved tool-use ability of an open-sourced model. ASearcher is able to process long trajectories and the work has developed efficient computing infra to leverage the computing resources. The performance of ASearcher is compared comprehensively on prevailing benchmark against a comprehensive set of baseline.

**Strengths:**

- Clear presentation and writing
- Comprehensive baselines
- Convincing results
- Effective and efficient approach

**Weaknesses:**

I don't have too much complaints about this work. The novelty of this work is moderate (in terms of using standard GRPO with fully async training) while I acknowledge the experimental efforts. I would suggest to open-source everything including the data filtering and all the components in the training. I have one main concern tho:

There are some components in the introduced method such as Dynamic Filtering and data filtering etc, e.g.,
```
 Finally, we filter out questions that are too hard for the modelor too easy for the model. Finally, from a total of 304k QA pairs, we retain 16k challenging samples
```
Maybe I missed, but I don't see ablation studies about them.

**Questions:**

NA

---

> ### Author Response · Authors · 2025-11-26
> **Response to Reviewer 1mWo**
>
> Thank you for your valuable feedback. We address your concerns below:
>
> ### W1: Open-source commitment
>
> We appreciate this suggestion and we have already released our code for training, evaluation, and data synthesis, as well as, the training data in [this anonymous link](https://anonymous.4open.science/r/ASearcher-7022). The code and data are also updated in the supplementary materials.
>
> ### W2: Missing ablation studies
>
> We have added comprehensive ablation studies to systematically evaluate the contribution of different components, directly addressing the reviewer's request. The ablation study starts from the Stage 1 checkpoint at the 200th training step, followed by an additional 200 steps of training under different settings to ensure a fair comparison.
>
> We highlight key conclusions from our ablation study,
> - **Training-time Turn Limit.** Training with a large turn limit significantly surpasses training with a small turn limit. This shows that the RL agent must be given the training budget (in terms of turns) to discover and reinforce long-horizon search strategies.
> - **Two-Stage Curriculum.** Compared with continued training with only Stage 1 data, the full training recipe with a two-stage curriculum helps the agent to learn stable and effective strategies by focusing on challenging queries that require at least 5 tool calls.
> - **Data Quality.** Finally, we also compare our training data with the baseline, AFM[1], by applying asynchrounus RL training using a 128 turn limit with training data from AFM. Training with AFM data is unable to incentivize complex search capabilities and achieves sub-optimal benchmark performance, showing that our RL data is of high quality.
>
> **Table A: Ablation Study**
>
> |  | # of Tool Calls at Training Time| GAIA | xBench |
> |---------------|---|------|--------|
> | **Full System** | | | |
> | ASearcher (Full) | 26.59 | 58.7 | 51.1 |
> | | | | |
> | **Ablating Training-time Turn Limit** | | | |
> | ASearcher w. Turn Limit=10 | 3.48 | 49.2 | 39.3 |
> | | | | |
> | **Ablating Training Data** |  | | | |
> | ASearcher w. Stage 1 Data Only | 5.40 | 51.6 | 43.0 |
> | ASearcher w. AFM Data | 4.12 | 50.9 | 39.9 |
>
> We believe that these clarifications and new results address your concerns and would be happy to provide any additional clarifications.
>
> [1] Li, W., Lin, J., Jiang, Z., Cao, J., Liu, X., Zhang, J., ... & Zhou, W. (2025). Chain-of-agents: End-to-end agent foundation models via multi-agent distillation and agentic rl. arXiv preprint arXiv:2508.13167.

---

> ### Author Response · Authors · 2025-11-28
>
> Dear Reviewer,
>
> I hope this message finds you well. As the discussion period is nearing its end, with **less than one week remaining**. I wanted to ensure that we have addressed your concerns satisfactorily. If there are any additional points or feedback you'd like us to consider, please let us know. Your insights are invaluable to us, and we're eager to address any remaining issues to improve our work.
>
> Thank you for your time and effort in reviewing our paper.

---

### Author Response · Authors · 2025-11-26
**Global Response**

We thank all reviewers for their detailed and constructive feedback. We apologize for the delay in our response, which was necessary to conduct **critical new experiments and comprehensive ablations** that directly address the core concerns raised by the reviewers.

**Major Revisions and New Evidence.** We have significantly strengthened our submission with the following new content,
- **Comprehensive Ablation Study** (Sec 4.4): We investigated the impact of using a large turn limit at training time and the two stage curriculum for data filtering process. We also make a comparison of training data between ASearcher and the AFM baseline through RL training.
- **Implementation Details** (Appendix C & D): We updated details of data synthesis, and prompts used by ASearcher.
- **Applying ASearcher Training on Qwen2.5-7B-Instruct** (Appendix E) We applied the training recipe on a smaller model, Qwen2.5-7B-Instruct, to show the scalability and effectiveness of our approach.
- **Evaluating WebSailor-32B Using DeepSeek-V3 Summary & TTS** (Appendix E): We evaluated WebSailor-32B using DeepSeek-V3 as the summary tool and using test-time search with $K$=16.
- **Code & Data Release** (Supplementary Materials): Our code for training, evaluation, and data synthesis, as well as the training data, are all released in [this anonymous link](https://anonymous.4open.science/r/ASearcher-7022) and supplementary materials. We commit to releasing the model upon acceptance.

**Key Contributions & Insights.** Our contributions are summarized as below,
- **RL-Enabled Emergence of Long-Horizon Strategy**: Our work uniquely demonstrates that **Reinforcement Learning enables the emergence of complex long-horizon search strategies** in an open-source model, including rigorous verification with more tool calls and nuanced analysis. This emergent capability is the primary source of our performance gain and aligns with the key insight of DeepSeek-R1 [1], where RL fully incentivizes reasoning in LLMs.
- **Full, Competitive Open-Source Recipe**: We present a full, end-to-end RL training recipe that achieves advanced performance on demanding benchmarks **without relying on commercial models**.
- Through additional ablation studies (Sec 4.4), we identify that **a large training-time turn limit** and **high-quality synthetic data** are the key to unlock this long-horizon search capability.

We believe these additions and clarifications could address the reviewer's concerns and highlight the novelty and impact of ASearcher.

[1] Guo, D., Yang, D., Zhang, H., Song, J., Wang, P., Zhu, Q., ... & Tan, Y. (2025). Deepseek-r1 incentivizes reasoning in llms through reinforcement learning. Nature, 645(8081), 633-638.

---

### Meta-Review · Area_Chair_Jhen · 2026-01-06

**Summary:**

This paper studies whether a single open-source model with reinforcement learning can acquire promising long-horizon search capabilities without relying on multiple commercial models. The authors combine large-scale synthetic data generation with large-scale long-horizon RL training under a very large turn limit (up to 128 actions), and employ a fully asynchronous agentic RL setup to make such long-horizon training efficient. Empirically, the resulting agent demonstrates strong emergent behaviors such as multi-step verification and uncertainty-aware reasoning, and achieves performance comparable to commercial models, e.g. OpenAI DeepResearcher and Kimi-Researcher.

Below, we summarize the main strengths identified in the reviews.

1. The proposed pipeline—comprising (a) a data generator to enable higher turn limits, (b) a two-stage curriculum, and (c) long-horizon reinforcement learning—represents a technically promising and well-motivated direction for scaling agentic reasoning.

2. Through a sequence of carefully designed technical components, the approach achieves competitive and in some cases stronger performance than commercial systems such as OpenAI DeepResearch and Kimi-Research, which is viewed as a notable empirical result.

**Reviewer Concerns:**

The main criticisms are that:

(1). Two reviewers (@HJhQ and @JUpp) raised concerns about the expensive cost of 16k H800 GPU hours.

(2). Reviewer @HJhQ criticizes the absence of a fully unified end-to-end pipeline.

(3). A few reviewers also raised concerns on the insufficient ablation studies:

         a) contribution of each component;

         b) aligning the usage of tool calls;

         c) aligning the use of smaller or non-reasoning models.

I appreciate the reviewers' hard work, however, considering the complexity of the tasks and the superiority of the empirical performance (even stronger than commercial models), the first concern of (1) this usage of16k H800 GPU hours is commensurate with the scale and ambition of the problem setting; (2) Given the complexity of the problem setting, it may not be necessary to be overly strict about whether the framework is fully unified or monolithic end-to-end.

After the rebuttal, the authors made non-trivial updates to conduct ablation studies in various aspects, which clearly demonstrate the necessity of each module, e.g. data synthesis, training-time turn limit, two-stage curriculum learning, in improving the reasoning abilities to the commercial level.

**Reviewer Scores:**

The paper is **boardline** in terms of review scores. 2 reviewers appreciate the competitive performance via synthetic-asynchronous RL training while 2 other reviewers raise concerns on the expensive cost and insufficient.

The review scores are 6, 4, 6, 2. The main rebuttal focused on the ablation studies of contributions in each component. After carefully reviewing the rebuttal process. I believe the authors have successfully addressed most of the concerns.

---

### Decision · Program_Chairs · 2026-01-26

Accept (Poster)